# Evaluation of suitable reference genes for qPCR normalisation of gene expression in a Achilles tendon injury model

Neil Marr[1]*, Richard Meeson[2], Richard J. Piercy[2], John C. W. Hildyard[1‡], Chavaunne T. Thorpe[1‡]

**1** Comparative Biomedical Sciences, Royal Veterinary College, London, United Kingdom, **2** Clinical Sciences and Services, Royal Veterinary College, Hatfield, United Kingdom

‡ JCWH and CTT are joint senior authors on this work.
* nmarr@rvc.ac.uk

**Data Availability Statement:** All relevant data are within the manuscript and its Supporting Information files.

## Abstract

Tendons are one of the major load-bearing tissues in the body; subjected to enormous peak stresses, and thus vulnerable to injury. Cellular responses to tendon injury are complex, involving inflammatory and repair components, with the latter employing both resident and recruited exogenous cell populations. Gene expression analyses are valuable tools for investigating tendon injury, allowing assessment of repair processes and pathological responses such as fibrosis, and permitting evaluation of therapeutic pharmacological interventions. Quantitative polymerase chain reaction (qPCR) is a commonly used approach for such studies, but data obtained by this method must be normalised to reference genes: genes known to be stably expressed between the experimental conditions investigated. Establishing suitable tendon injury reference genes is thus essential. Accordingly we investigated mRNA expression stability in a rat model of tendon injury, comparing both injured and uninjured tendons, and the effects of rapamycin treatment, at 1 and 3 weeks post injury. We used 11 candidate genes (*18S*, *ACTB*, *AP3D1*, *B2M*, *CSNK2A2*, *GAPDH*, *HPRT1*, *PAK1IP1*, *RPL13a*, *SDHA*, *UBC*) and assessed stability via four complementary algorithms (Bestkeeper, deltaCt, geNorm, Normfinder). Our results suggests that *ACTB*, *CSNK2A2*, *HPRT1* and *PAK1IP1* are all stably expressed in tendon, regardless of injury or drug treatment: any three of these would serve as universally suitable reference gene panel for normalizing qPCR expression data in the rat tendon injury model. We also reveal *18S*, *UBC*, *GAPDH*, and *SDHA* as consistently poor scoring candidates (with the latter two exhibiting rapamycin- and injury-associated changes, respectively): these genes should be avoided.

## Introduction

Tendons are collagen-rich tissues connecting muscle to bone and can experience extremely high stresses and strains during locomotory activities such as running and jumping, making them prone to injury [1, 2]. Injury results in disruption to the tendon extracellular matrix (ECM), leading to pain and loss of function [3]. Healing occurs in several phases, and is

**Funding:** Funded by Versus Arthritis (22607). The funders had no role in study design, data collection and analysis, decision to publish, or preparation of the manuscript.

**Competing interests:** The authors have declared that no competing interests exist.

mediated by both endogenous and exogenous cell populations; however repair is often insufficient, with the formation of functionally inferior fibrotic scar tissue leading to chronic tendinopathy and a high rate of re-injury [4–6]. While it is evident that tendons house many cell types, including several populations of tenocytes, mural cells, endothelial cells and immune cells [7–9], the roles of specific populations in tendon healing remain incompletely understood.

Several pre-clinical animal models of tendon injury exist, which use non-surgical and surgical techniques depending on suitability and translation [10, 11]. Rodents, particularly the rat (Rattus norvegicus) offer a practical and cost-effective model system, and rat tendons, particularly the Achilles tendon, exhibit structural functional, and cellular similarities to those in humans, making them valuable models for understanding *in vivo* dynamics of tendon repair and assessing effectiveness of therapeutics [12, 13]. Assessment outcomes of animal models rely on robust qualitative methodologies, such as histology or immunolabelling, and quantitative approaches such as protein and gene expression, to determine the extent of tendon repair and potential contributions from therapeutic interventions.

Quantitative polymerase chain reaction (qPCR) is a highly sensitive technique for measuring levels of target nucleotide sequences: use of qPCR with cDNA allows changes in gene expression at the transcriptional level to be determined with high accuracy, but such gene expression data must first be normalised, to account for innate variation in RNA isolation/ integrity and cDNA synthesis efficiency. Normalisation typically uses reference genes (RG): genes known to exhibit stable expression between the conditions tested, and frequently used RGs include *18S* ribosomal RNA, β-actin (*ACTB*) or glyceraldehyde 3-phosphate dehydrogenase (*GAPDH*), with the latter two chosen primarily due to their ostensible stability at the protein level. Historically, studies have relied on a single reference gene for normalisation. It is increasingly clear, however, that use of more than one reference is essential for good normalisation (as per MIQE guidelines [14]), and furthermore that no 'universal' reference genes exist (indeed several studies, including from the authors, have demonstrated that *ACTB* and *GAPDH* should be actively avoided under some conditions [15]): instead, reference genes appropriate for the experimental condition should be determined empirically. Multiple computational approaches have been developed to determine appropriate reference genes, with BestKeeper [16], ΔCt (herein deltaCT) [17], geNorm [18], and Normfinder [19] being popular choices. Each approach requires a dataset of gene expression values for a number of candidate references, measured in multiple samples representative of the experimental conditions, but each approach also assesses expression stability in a subtly different fashion, and resultant scores are thus not always identical: genes scoring highly by multiple independent assessment are likely to represent very strong references, but differences in scoring can also provide insights into underlying biology.

Identifying the most stable RG candidates for normalisation of expression is essential for determining transcriptional changes in tendons. Given that our work has previously identified vascular cell subpopulations within tendons across species [20, 21], dynamic changes in gene expression relating to angiogenesis, inflammation and ECM remodelling alone demonstrated the necessity for robust normalisation procedures when studying tendon injury in rodent models. We have previously published a repeatable, robust, minimally-invasive needle-induced injury model in rodents to investigate tendon healing, identifying gross tissue-wide, morphological and cellular changes in response to injury taking place from 7d to 21d of surgery [22]. However, gene expression was not surveyed: given the dynamic changes in tissue morphology and resident cell populations we observe, a robust normalisation procedure for gene expression analyses is required to further validate our surgical model of tendon injury.

We have thus used four reference gene determination packages (BestKeeper, DeltaCT, GeNorm and Normfinder) to identify genes appropriate for normalizing gene expression in

our rat tendon injury model. As all methods benefit from substantial sample cohorts and candidate panels, our study used a sample collection including both injured and uninjured tendons, collected at 1 and 3 weeks post-injury, with and without concurrent therapeutic rapamycin administration (35 samples in total, N = 3–5 per group). Similarly, we used a panel of 11 reference candidates, involved in various pathways including protein synthesis, cell metabolism and cytoskeletal elements (*18S*, *ACTB*, *AP3D1*, *B2M*, *CSNK2A2*, *GAPDH*, *HPRT1*, *PAK1IP1*, *RPL13a*, *SDHA*, *UBC*). Several of these genes have been shown by us and others to be consistently high scoring across several comparative scenarios, both across animal models and modes of musculoskeletal experimentation including *in vitro* cell culture [23–25] and *in vivo* disease pathology [26–29]. Our dataset allows us to both identify the strongest reference candidates overall, but also to determine if stronger but condition-specific references exist (indicative of injury- or drug-sensitive expression).

## Materials and methods

### Ethical statement

This study and all-inclusive procedures complied with the Animals (Scientific Procedures) Act 1986, approved by the Royal Veterinary College Animal Welfare and Ethical Review Body (ID:2016-0096N; June 2017), under Home Office project license PB78F43EE (license holder: CTT), and reported according to ARRIVE guidelines.

### Housing and husbandry

Animals were housed in groups of 3 in individually ventilated polypropylene cages, subjected to 12h:12h LD cycles between 08:00 and 20:00 at a temperature of 21˚C. Animals were fed *ad libitum* on a maintenance diet (Special Diet Services, Chelmsford, UK), and provided with a rotational enrichment programme.

### Surgical procedures and drug administration

Female Wistar rats (n = 24 total; 12 weeks old; weight = 206g, range = 141-226g) had the needle-induced Achilles tendon injury procedure performed under general anaesthesia (isoflurane; 2.5–3%), as previously described [22]. Contralateral hindlimbs were untreated as controls. Pre- and post-operative analgesia was provided (0.05 mg/kg buprenorphine, sub-cutaneous, for 48 h post-operatively). In the three days post-surgery, behaviour was scored according to pre-defined criteria, assessing weight, appearance, lameness, unprovoked behaviour, body condition and respiration to ensure that any suffering was minimised. Rapamycin treatment groups (n = 12) were injected intraperitoneally (i.p.) with 2 mg/kg rapamycin, and control treatment groups (n = 12) were injected with volume-matched vehicle solution that consisted of 5% PEG and 5% TWEEN® 80 in sterile water. Rats were euthanised 7 (n = 12) or 21 days (n = 12) post-injury induction. Dosing regimens were started 24 h post-operation and ended 24h prior to euthanasia. For the day 7 groups, animals received a total of 5 i.p. injections, whereas day 21 groups received a total of 19 i.p. injections.

### Tissue collection and processing

Rats were euthanised by a rising concentration of $CO_2$ and confirmed by cervical dislocation and cardiac puncture. Both Achilles tendons were harvested immediately and mounted in OCT embedding matrix (Cell Path, Newtown, UK), then snap-frozen in dry ice-chilled hexane and stored at −80˚C.

## RNA isolation

RNA was isolated from cryosections of OCT-embedded tendons (30–50 sections, 12 μm thickness), collected serially to those mounted for histology. Isolations used 1–1.5 mL TRIzol™ as per manufacturer's guidelines, with inclusion of an additional 1:1 chloroform extraction following phase separation. To maximise yields, isopropanol precipitations were supplemented with 10μg glycogen, and 50μl (0.1 vols) 3M Sodium acetate pH 5.5 (Ambion). RNA quality was assessed using a DS-11 Spectrophotometer (DeNovix): any samples with 260/230 absorbance ratios lower than 1.7 were cleaned via a second isopropanol precipitation.

## cDNA synthesis

cDNA synthesis (800 ng RNA per 20ul reaction) was conducted using High-Capacity cDNA synthesis kit (ThermoFisher), using oligodT and random priming. Reactions were then diluted 1:10 with nuclease-free water (Qiagen) and stored at -20˚C until needed.

## RNA/cDNA quality control

Tendon tissues (particularly under normal conditions) are only modestly transcriptionally active, and the highly fibrous nature of the tendon environment further renders RNA isolation challenging. Therefore, to preserve quality, a stringent rejection criterion was applied during sample collection: any RNA samples with insufficient yields for cDNA synthesis were excluded, and following cDNA synthesis any samples with aberrantly high Cq values (indicative of failed cDNA synthesis or underlying sample degradation) were similarly excluded. Our final dataset thus consisted of 35 samples, with N values of 3–5 per sample group (see **Table 1**).

## qPCR

qPCR reactions were performed on a CFX384 thermal lightcycler in 384-well plates (white hard-shell thin wall, BioRad) using PrecisionPLUS SYBR green mastermix (Primerdesign). 10ul PCR reactions were carried out in triplicate using 2 μL diluted cDNA (ca. 8 ng cDNA assuming 1:1 synthesis) per well. All runs included melt curve analysis as standard and template-free controls to confirm single amplicons and primer specificity with amplification. Quantification cycle (Cq) values were determined by regression and converted to (linear) Relative Quantities (RQ) where appropriate (see below).

## Reference gene selection and primer details

Candidate reference genes (RGs) were selected based on prior performance in previously published studies [23, 26–29]. All primers (described in **Table 2**) were either taken from the literature (where indicated) or designed using Primer3 [30].

**Table 1. Data subsets.**

| Subset | Details | Sample number (n) |
|---|---|---|
| All data | All animals, all limbs, all treatments, all timepoints | 35 |
| All uninjured | All contralateral control limbs only | 19 |
| All injured | All injured limbs only | 16 |
| All rapamycin | All animals (control and injured limbs), all rapamycin treated only | 18 |
| All vehicle | All animals (control and injured limbs), all vehicle treated only | 17 |

**Table 2. Candidate genes and primer sequences.**

| Gene symbol | Gene name | Primer sequence (5'-3') | | Reference |
|---|---|---|---|---|
| *18S* | *18S* ribosomal RNA | Forward: AAACGGCTACCACATCCAAG | | [31] |
| | | Reverse: TTGCCCTCCAATGGATCCT | | |
| *ACTB* | Actin Beta | Forward: ATTGCTGACAGGATGCAGAA | | [31] |
| | | Reverse: TAGAGCCACCAATCCACACAG | | |
| *AP3D1* | Adaptor Related Protein Complex 3 Delta 1 Subunit | Forward: AACATGGAACTCAACGTGCT | | Designed |
| | | Reverse: GGTGGCTCCTTCATCATCCT | | |
| *B2M* | Beta-2-Microglobulin | Forward: ACATCCTGGCTCACACTGAA | | [31] |
| | | Reverse: ATGTCTCGGTCCCAGGTG | | |
| *CSNK2A2* | Casein Kinase 2 Alpha 2 | Forward: TCCATGGGCAGGACAACTAT | | Designed |
| | | Reverse: AAAGTTTTCCCAGCGCTTCC | | |
| *GAPDH* | Glyceraldehyde-3-Phosphate Dehydrogenase | Forward: CTGACATGCCGCCTGGAGA | | [32] |
| | | Reverse: ATGTAGGCCATGAGGTCCAC | | |
| *HPRT1* | Hypoxanthine Phosphoribosyltransferase 1 | Forward: GCTGAAGATTTGGAAAAGGTG | | [31] |
| | | Reverse: AATCCAGCAGGTCAGCAAAG | | |
| *PAK1IP1* | PAK1 Interacting Protein 1 | Forward: TGTGATACCCTAGTGTGCCTC | | Designed |
| | | Reverse: CCTTCTCATCTGTCCCTCGG | | |
| *RPL13A* | Ribosomal Protein L13a | Forward: GGATCCCTCCACCCTATGACA | | [33] |
| | | Reverse: CTGGTACTTCCACCCGACCTC | | |
| *SDHA* | Succinate Dehydrogenase Subunit A | Forward: AGACGTTTGACAGGGGAATG | | [31] |
| | | Reverse: TCATCAATCCGCACCTTGTA | | |
| *UBC* | Ubiquitin C | Forward: ATGTCGAGCCCAGTGTTAAC | | Designed |
| | | Reverse: TGCAATGAAACTTGTTAACAGCT | | |

## Data analysis

BestKeeper, geNorm, deltaCt and normfinder analyses were performed on the entire dataset as well as subsets of the data as per **Table 2**. Bestkeeper and deltaCt used raw Cq values, while geNorm and normfinder used RQ values. The normalisation factor (NF) used for subsequent validation was the (per-sample) geometric mean of the RQ values of the three highest scoring candidates: as RQ values are linear, normalisation was conducted conventionally (division by NF) and data was subsequently log-transformed for analysis. All data analysis was performed using Microsoft Excel and data was visualised using GraphPad Prism version 10.2.2 for Windows (GraphPad Software, Boston, Massachusetts USA).

## Results

### Distribution of Cq values across dataset

Raw Cq values (all samples, all candidate genes) act as a first-pass validation and assessment of the reference candidate panel. Variable expression was observed in *18S* and *GAPDH*, whilst remaining candidates were more consistent (**Fig 1**). Across the entire candidate panel, Cq values spanned a broad range in line with expected abundance, with the highest expression found in *18S* as expected, abundant expression found in *RPL13a*, *ACTB* and *B2M*, and more modest expression across all other candidates. For several genes, expression within uninjured samples appeared slightly lower than within injured, indicative of potentially damage-associated expression differences.

# Raw Cq

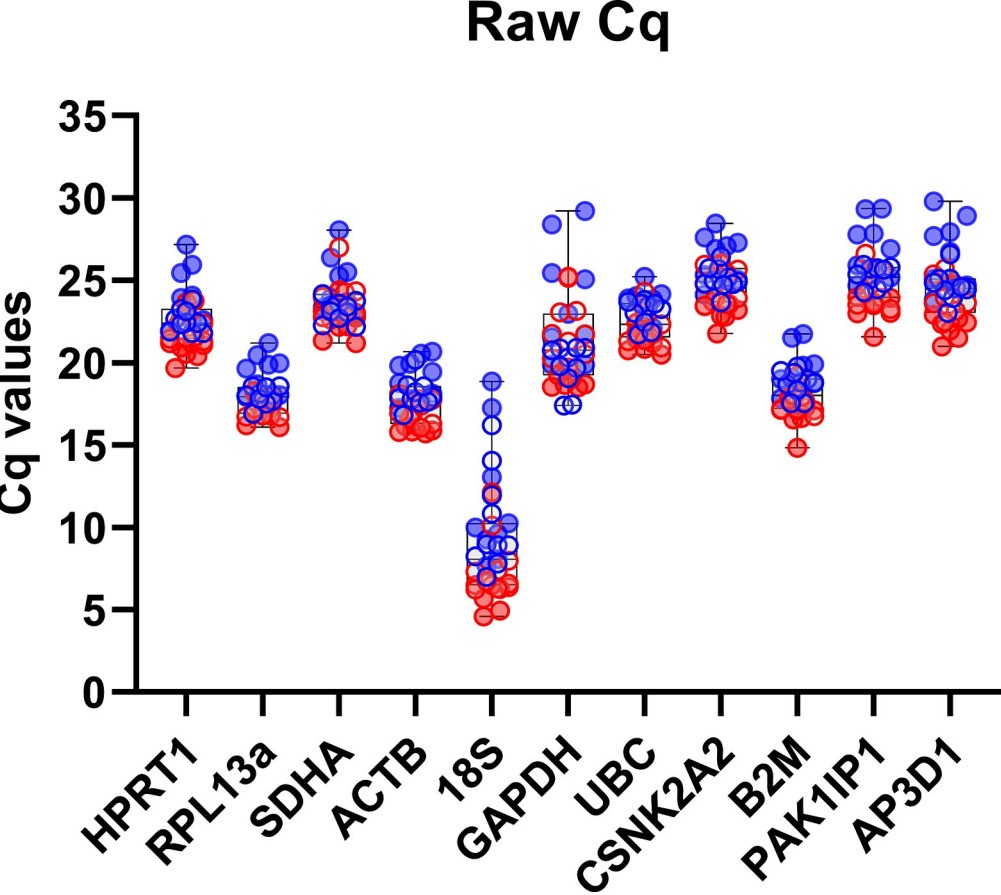

**Fig 1. Raw quantification cycle (Cq) values for each candidate reference gene.** Box-and-Whisker plot featuring minimum and maximum Cq values for each RG. Injured tendon = red, control tendons = blue, rapamycin = open circles, vehicle = filled circles. Note: lower Cq values indicate greater gene expression.

## BestKeeper analyses

BestKeeper analysis averages expression across all candidate genes to create a consensus expression profile reflecting the mean behaviour of the entire dataset. Individual genes are then ranked by their correlation coefficient (Pearson; r) with this consensus 'best keeper'. Analysis of our full dataset (**Fig 2A**) revealed six genes (*CSNK2A2*, *ACTB*, *HPRT1*, *RPL13a*, *AP3D1* and *PAK1IP1*) as strong candidates, with near identical correlation coefficients (0.93–0.96). These genes continued to score highly when data was analysed as specific subsets: uninjured or injured samples (**Fig 2B and 2C**), and drug- or vehicle-treatment (**Fig 2D and 2E**). Indeed strong positive correlations (>0.8) were observed for the majority of our candidate panel, though *B2M*, *UBC*, *18S*, *SDHA* and *GAPDH* typically scored more poorly than other candidates (with the latter two performing markedly less well in rapamycin treated samples alone; **Fig 2D**).

## GeNorm analyses

GeNorm approaches candidate selection via iterative pairwise comparisons. The pairwise variation of each individual candidate with all others is summed (termed the stability value, M), and the gene with the highest M (i.e. the most variable) is discarded. The process is then

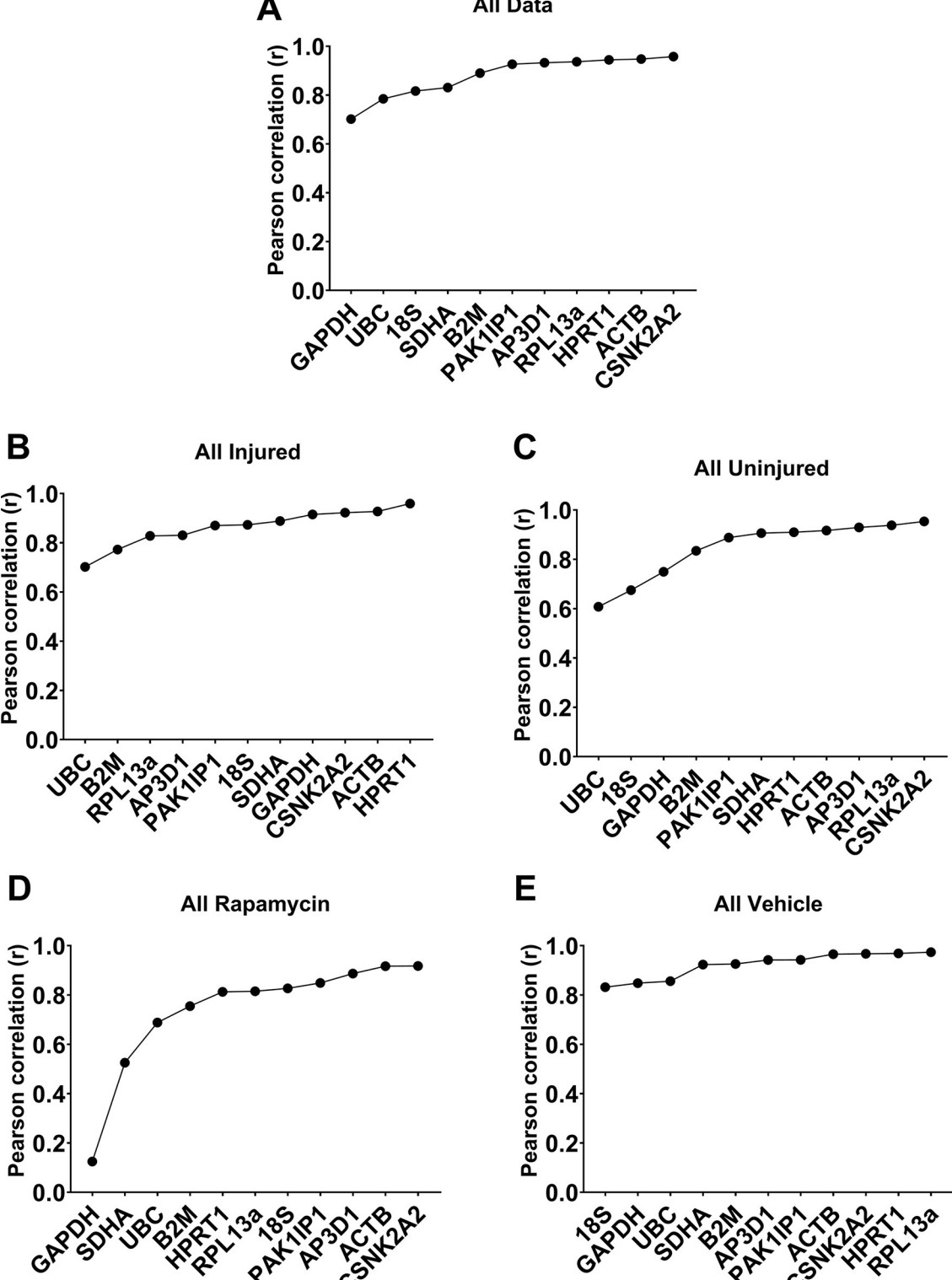

**Fig 2. Rankings produced from BestKeeper analyses of 11 candidate reference genes.** Coefficient of correlation (r) values for the complete dataset (**A**), or sample subsets (all injured (**B**), all uninjured (**C**), all rapamycin (**D**) and all vehicle treated (**E**) samples). Genes are ranked (left to right) from least stable (low correlation) to most stable (high correlation).

repeated until only a single pair of highly correlated genes remain (the 'best pair'), with all other candidates ranked by their final M values. The underlying assumption is that genes from unrelated categories, but which nevertheless appear highly correlated, are likely to reflect overall cDNA content. GeNorm assessment of our dataset (**Fig 3**) largely agreed with Bestkeeper: *18S*, *GAPDH*, *SDHA* and *UBC* were near-consistently ranked last, while *RPL13a*, *CSNK2A2*, *HPRT1*, *ACTB* and *PAK1IP1* scored highly overall (**Fig 3A**) and in injured (**Fig 3B**), uninjured (**Fig 3C**), rapamycin-treated (**Fig 3D**), and vehicle-treated (**Fig 3E**) subsets (indeed *CSNK2A2* formed one half of the best pair in all assessments except the 'vehicle-only' subset in **Fig 3D**). By convention, M values <0.5 are considered acceptable as references: none of our candidates achieved this threshold when the dataset was assessed as a whole (**Fig 3A**). However, M values were typically lower within data subsets (achieving sub 0.5 M values in several cases; **Fig 3B–3E**), indicating that expression was substantially more stable within groups than between groups (as might be expected): the higher M values for our complete dataset thus presumably reflects this variation (we note that for transcriptionally plastic scenarios investigators often accept M <1.0). In addition to determining the best pair of genes, geNorm also determines changes in pairwise variation (i.e. gains or losses in normalisation stability) obtained by use of additional RGs. In all cases, use of three (rather than two) RGs resulted in modest gains in stability, but use of the best pair alone was nevertheless sufficient (variation < 0.2; **S1 Table**).

## Normfinder analyses

We next employed Normfinder analysis. Unlike the previous approaches, which utilise pairwise comparisons, Normfinder assesses stability of individual genes independently across datasets. This can be conducted ungrouped (i.e. overall stability) or grouped (assessing stability across user-defined categories): use of the latter analysis is helpful for identifying genes which might appear ostensibly stable but which also exhibit modest but consistent group-specific behaviour. Ungrouped analysis of our entire dataset largely agreed with the findings above: *ACTB*, *CSNK2A2*, *HPRT1* and *PAK1IP1* were the strongest candidates (**Fig 4A**), and ungrouped analyses of data subsets (**S1 Fig**) similarly supported this, with these four genes consistently scoring highly. We then turned to grouped analysis, using injured/uninjured (**Fig 4B**), drug/vehicle treatment (**Fig 4C**), timepoint (**Fig 4D**), or the combination of all three factors as groups (**Fig 4E**). These four RGs exhibited consistently strong performance regardless of grouping, suggesting that they are indeed very strong candidates. Grouped analysis also suggests a best pair: a pair of genes with greater combined stability than any individual gene. In all cases the best pair was formed by two of these four RGs. Interestingly, *B2M* scored markedly higher in samples grouped by injury than in other groupings (**Fig 4B**), suggesting that expression of this gene might be insensitive to injury, but vary with treatment and time (**Fig 4C and 4D**). Lastly, as with geNorm and Bestkeeper: *UBC*, *GAPDH* and *18S* were shown to be poor candidates under both ungrouped and grouped analyses, with the latter two being particularly poorly stable under essentially all scenarios.

## DeltaCt (ΔCt) analyses

The DeltaCT method was the fourth approach employed to analyse our candidate reference genes. This method measures differences in Cq value (the deltaCt, or dCt) of all possible gene pairs on a per-sample basis, then determines the standard deviation (SD) of these dCt values: genes which vary in a consistent manner will exhibit consistent per-sample dCt, and thus low SD, while highly variable genes will exhibit higher SDs. Mean dCt SD values are thus used to rank candidates. Under DeltaCt analysis (**Fig 5**), *CSNK2A2*, *HPRT1*, *ACTB* and *PAK1IP1* again scored consistently well overall (**Fig 5A**), in injured (**Fig 5B**) or uninjured tendons

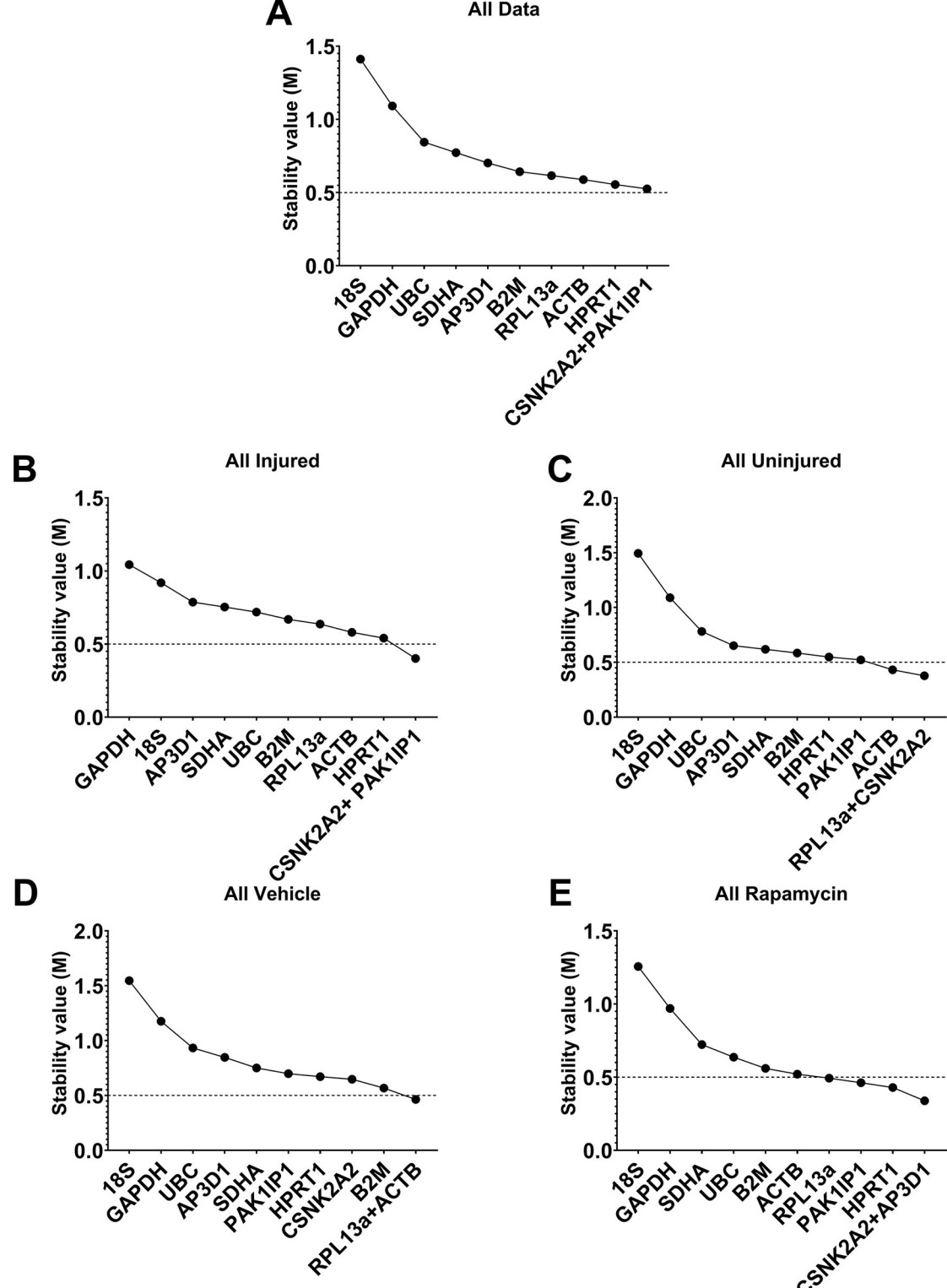

**Fig 3. Representative outputs of GeNorm analyses of 11 candidate reference genes.** RGs are ranked by average expression stability (M) from left to right (least to most stable) for all data (**A**), injured (**B**), uninjured (**C**), rapamycin (**D**) and vehicle (**E**) treated samples (as indicated). Dashed line (M = 0.5) = conventional threshold for acceptable RG candidates (lower M values).

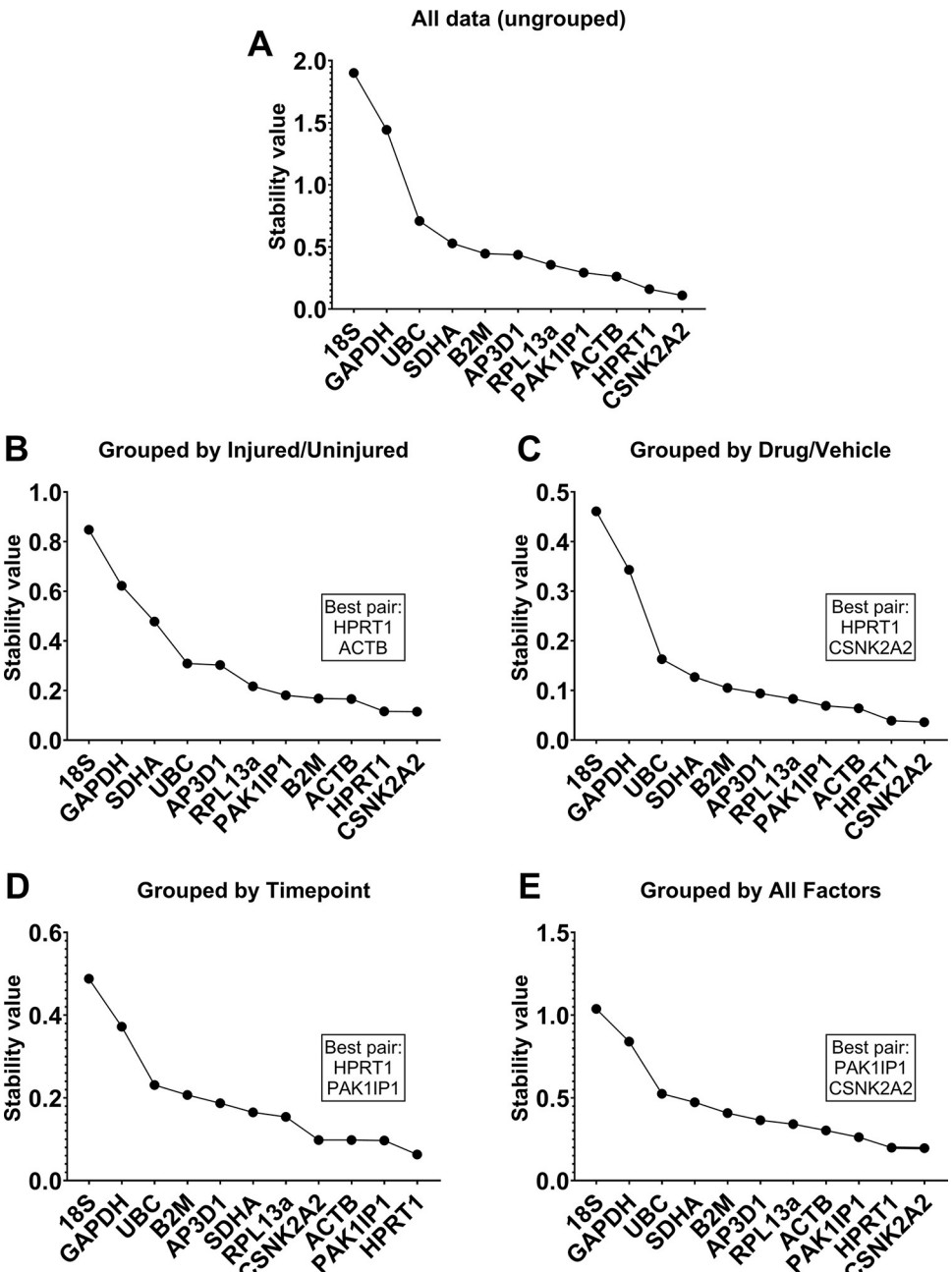

**Fig 4. NormFinder analyses.** Rankings for 11 candidate RGs computed by Normfinder using ungrouped analyses of the entire dataset (**A**), and grouped analyses by injury (**B**), drug treatment (**C**), timepoint (**D**) and by all factors (**E**). Lower scores correspond to higher stability. Best pair combinations are indicated in boxes respectively.

(**Fig 5C**), and in both rapamycin (**Fig 5D**) and vehicle (**Fig 5E**) treatment. *18S*, *GAPDH* and *UBC* invariably performed poorly, ranking in the lowest four genes in all cases.

## Integrated analyses

All four methods used here consistently suggested that *ACTB*, *CSNK2A2*, *HPRT1* and *PAK1IP1* represent strong RGs. Similarly, *18S*, *GAPDH*, and *UBC* were near-invariably ranked

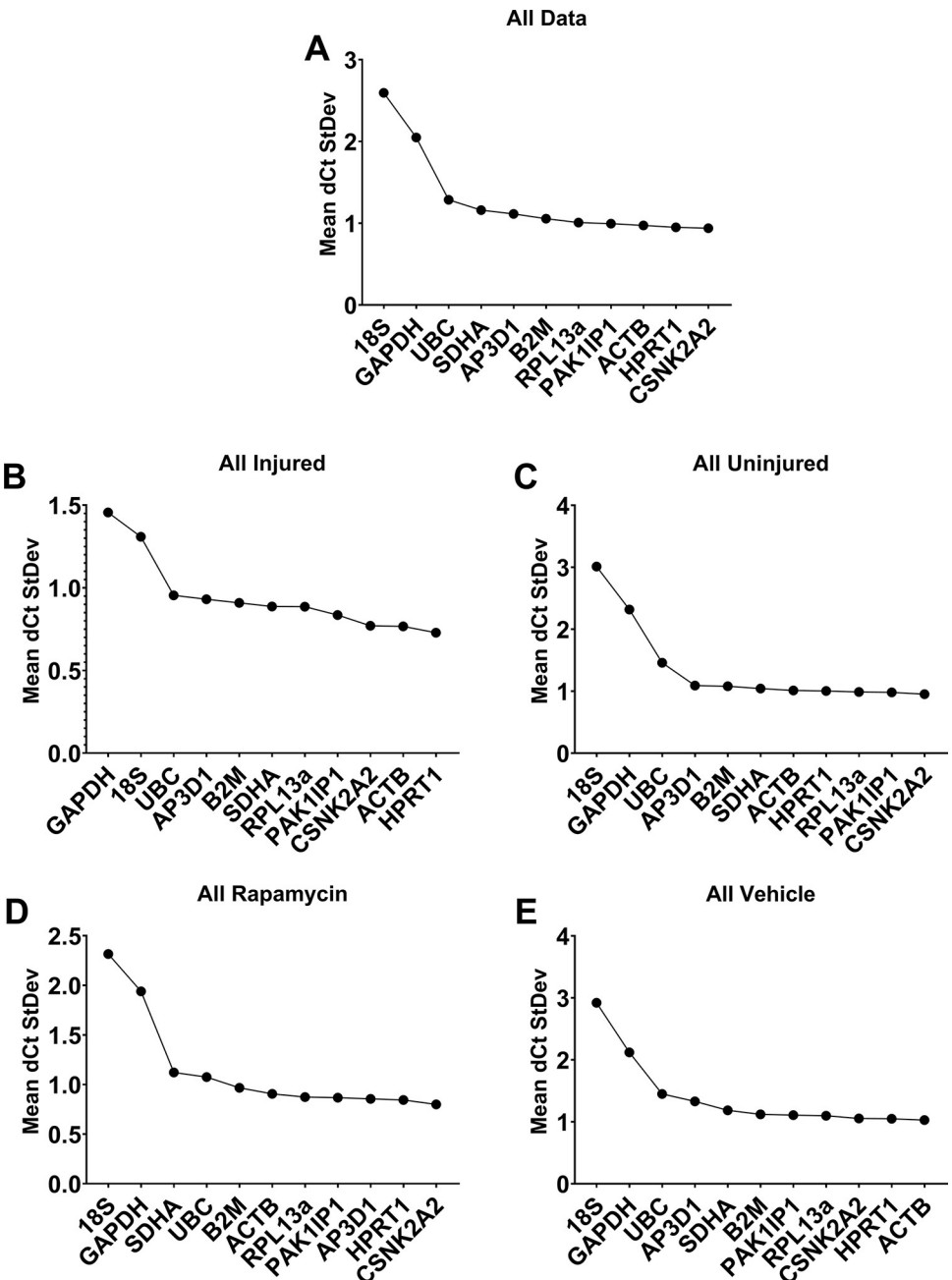

**Fig 5. ΔCt rankings.** Ranked scores of 11 candidate RGs assessed by DeltaCT method across all data (**A**), all injured tendons (**B**), all uninjured tendons (**C**), all rapamycin treated rats (**D**) and vehicle treated rats (**E**). Lower ΔCt scores corresponds to higher stability.

last, suggesting that these three genes are wholly unsuited for normalisation in the rat tendon injury model. To clarify this explicitly, we employed an integrated ranking system similar to that reported recently [34], calculating the geometric mean score of each gene across the four methods. Our integrated analyses (**Fig 6**) confirmed that *18S*, *GAPDH* and *UBC* were the least stable not only in our full dataset (**Fig 6A**), but also all four data subsets: injured tendon

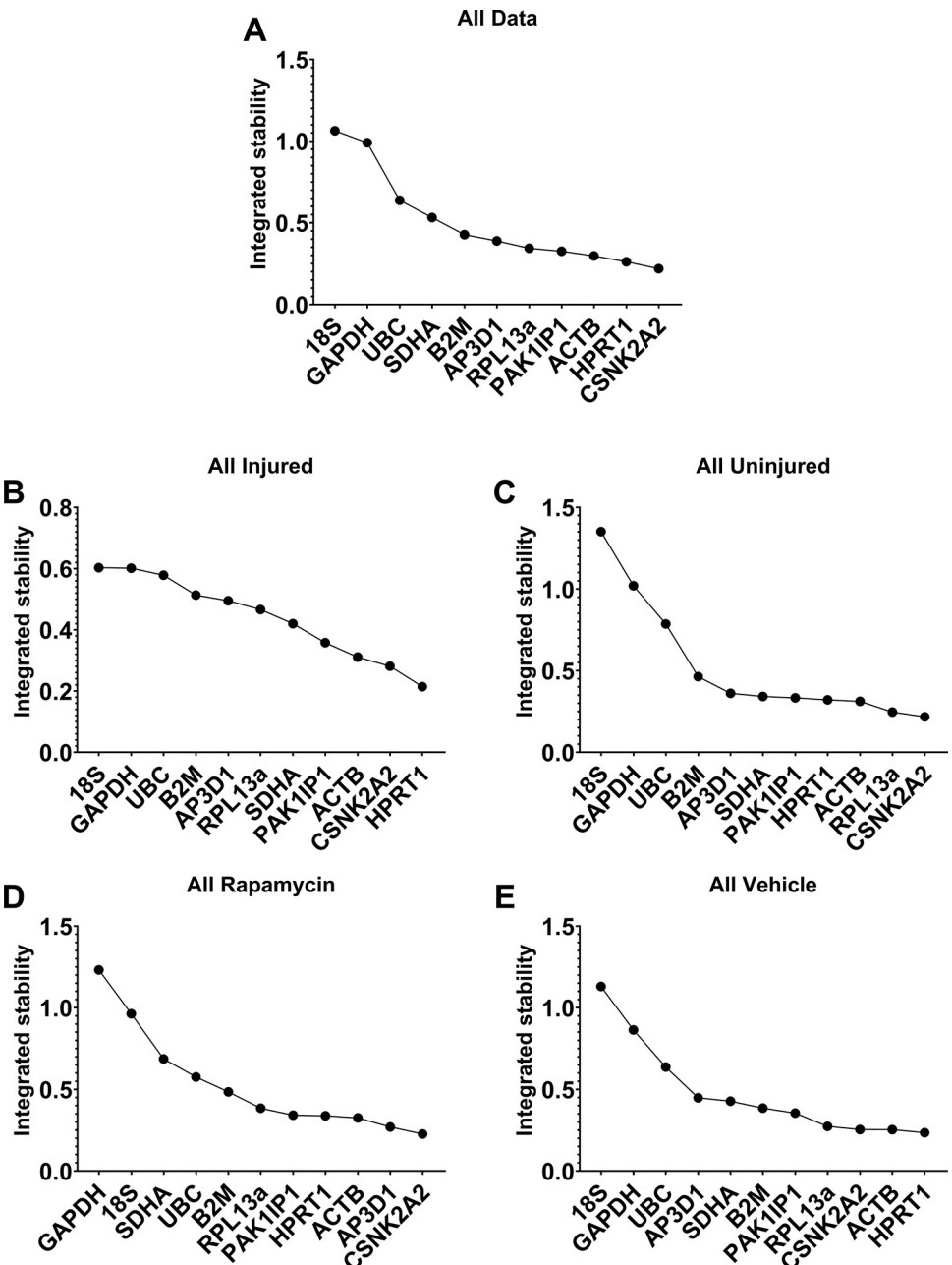

**Fig 6. Integrated rankings of BestKeeper, DeltaCT, geNorm and Normfinder analyses.** Geometric mean ranks from all four algorithms assessed as a whole dataset (**A**), injured tendons only (**B**), uninjured tendons only (**C**), rapamycin treatment only (**D**), vehicle treatment only (**E**).

(**Fig 6B**), uninjured tendon (**Fig 6C**), rapamycin treatment (**Fig 6D**) and vehicle treatment (**Fig 6E**). Aggregate scores also confirmed the suitability of *PAK1IP1*, *CSNK2A2*, *ACTB*, and *HPRT1* as reference genes (*RPL13a* also ranked comparatively highly, particularly in uninjured and vehicle-treated groups, suggesting that this might be a suitable RG for healthy tendon, but less suited to injury or pharmacological intervention scenarios).

## Validation of normalisation factor

To validate our findings, we used an approach we have employed previously [26, 27, 35], using the highest ranking candidates (*ACTB*, *CSNK2A2*, and *HPRT1*) to normalise the lowest: consistently poor scoring for a given gene might imply significant condition-specific expression, something that normalisation to strong RGs should reveal. For this we examined two genes, *GAPDH* and *SDHA* (**Fig 7**): the former was selected on basis of uniformly poor scoring, while the latter was chosen because this gene typically exhibited higher rankings in specific subsets than in the complete dataset (indicative of between-group variation). In comparison to non-normalised data, normalisation of *SDHA* (**Fig 7A and 7B**) or *GAPDH* (**Fig 7C and 7D**) to our three-gene normalisation factor (NF) reduced within-group coefficient of variation (CoV) -an outcome expected from good normalisation- and revealed distinct group-associated expression patterns. Expression of *SDHA* within individual groups was comparatively consistent (explaining the higher rankings in group-specific subsets), yet demonstrated clear injury-associated reductions that were potentially further exacerbated by rapamycin treatment. *GAPDH* conversely appeared both inherently variable within groups (even after normalisation), but also differentially responsive to rapamycin treatment: in drug-treated tendon samples, mean expression appeared lower in the presence of injury, but higher in the absence of injury. These findings explain the ranking of these two genes within our dataset, and also demonstrate the utility of a normalisation factor derived from *ACTB*, *CSNK2A2*, and *HPRT1* in the rat tendon injury model.

## Discussion

Tendon injuries present with a dynamic pathology; whether chronic or acute, most injuries will result in disruption of the native ECM and subsequently losses in mechanical integrity, followed by activation of inflammatory pathways, recruitment of endogenous and exogenous cell populations, and neovascularization, with concomitant alteration of mechanosensitive gene expression [5, 36, 37]. This diverse and dynamic transcriptional environment potentially renders normalisation of expression data challenging, however as we demonstrate here in a rat tendon injury model, universally suitable reference genes can nevertheless be identified. In all analyses, four of our 11 genes (*ACTB*, *CSNK2A2*, *HPRT1*, *PAK1IP1*) were consistently high ranking, suggesting that these four represent strong references, regardless of injury or drug treatment. The precise order in which these genes were ranked did differ across the different methods and comparative scenarios, yet all were nevertheless highly comparable, with little evidence to favour one over another. As per the MIQE guidelines, use of a single RG is highly discouraged: two reference genes represent the minimum. Yet, use of four references could be considered excessive, and we thus elected to choose three of these four: integrating our findings to generate aggregate rankings (**Fig 6**) placed *ACTB*, *CSNK2A2*, and *HPRT1* as the highest ranking overall. Our validation studies using these three candidates (**Fig 7**) suggests that these are indeed highly suited to normalising gene expression in this model.

 Literature on reference gene studies in the tendon is limited, however our findings are nevertheless in agreement with some other studies: *ACTB* and *HPRT1* were determined as universal reference genes in human tendon injuries both for the ACL [29] and rotator cuff [28]. We stress that we cannot necessarily assert that our top ranking RGs here are directly applicable to other species: for example, we have previously shown that *ACTB* is a strong reference in both healthy and dystrophic mouse skeletal muscle, but not canine muscle, while the reverse is true for *HPRT1* [26, 27]. For tendon specifically, the tissue architecture, peak stresses and gross physical dimensions vary dramatically from mouse to horse, and thus a degree of species-

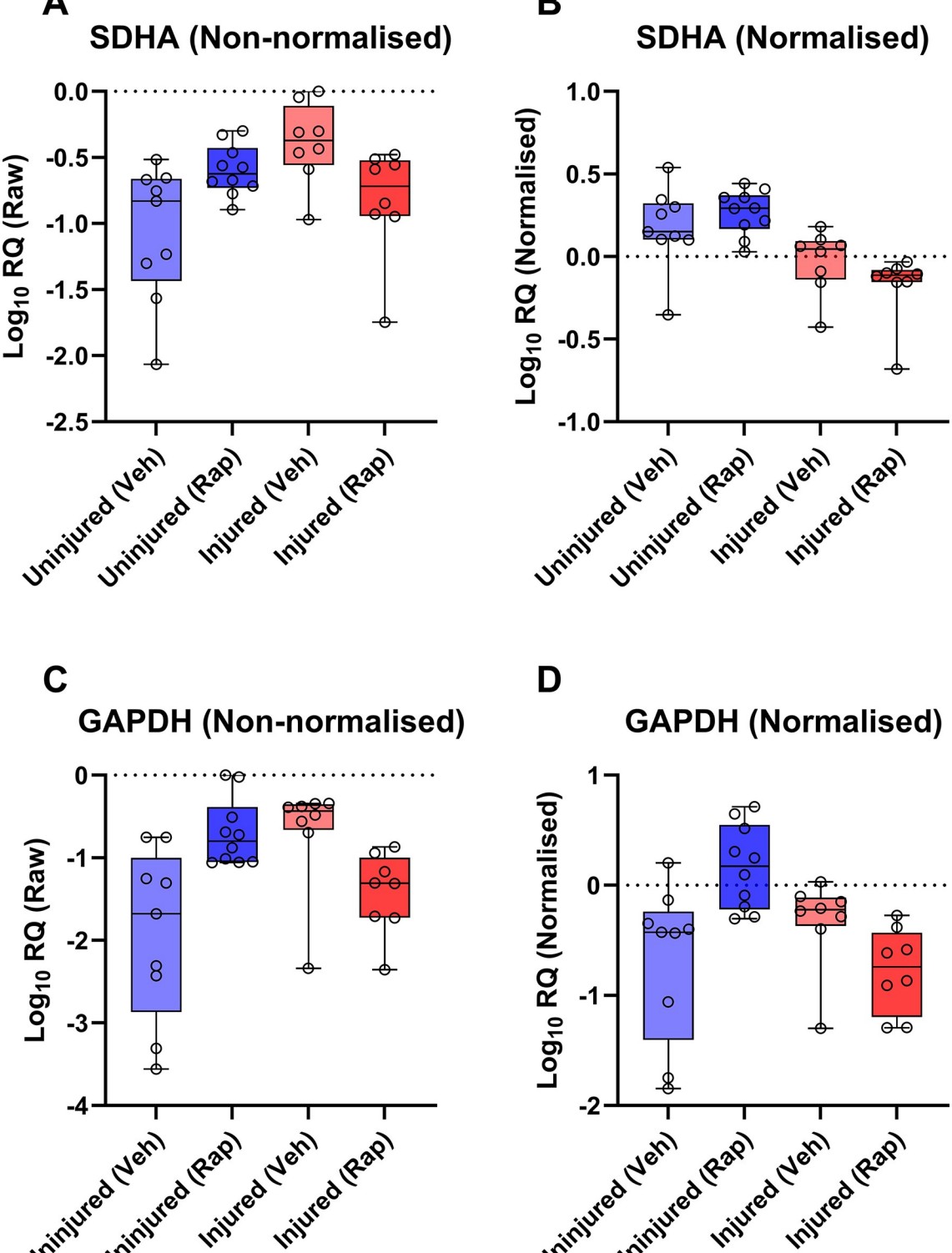

**Fig 7. Validation of high-ranking RGs.** Log RQ values for raw data (**A** & **C**) and 3-gene (*ACTB*, *HPRT1* and *CSNK2A2*) normalised data (**B** & **D**) for the low-ranking RGs *SDHA* (top) and *GAPDH* (bottom). Normalisation reduces within-group variability for both genes, and reveals injury-associated changes in *SDHA*, and injury-sensitive rapamycin-associated changes in *GAPDH*.

specific variation is to be expected. One previous study identified two of our poorest candidates, *SDHA* and *GAPDH*, as high-performing RGs in uninjured equine tendon tissues (and *ACTB* and *GAPDH* in injured equine SDFT) [38], though we note the authors only tested four candidate references, potentially limiting the breadth of their interpretations.

Across our analyses, several genes consistently performed poorly, with *18S* and *GAPDH* in particular almost invariably ranking last (typically by a substantial margin). *18S* has substantial pedigree as a reference, but suffers from several shortcomings that make it a questionable choice: as an rRNA, it lacks a polyA tail (necessitating random priming for cDNA synthesis, and precluding its use in oligodT-primed samples), and as a component of the ribosome it is also one of the most abundant RNA species within any given total RNA sample, being consequently effectively a metric for total RNA content rather than mRNA content. In our dataset *18S* was demonstrably highly variable between samples, and also appeared to be injury-associated (**Fig 1**), possibly reflecting the higher cellular diversity of injured tendons, rather than transcriptional activity *per se*. *GAPDH* similarly has established tenure as a reference, but as we and others have shown (and further show here), this gene is not appropriate under many circumstances, and exhibits considerable expression plasticity. Previous studies have suggested *GAPDH* expression in rodent tendons changes even with exercise [39]; as a core component of the cellular glycolysis chain, this is perhaps not unexpected. Our data here suggests that expression of *GAPDH* does not strongly differ between injured/uninjured tendons in the absence of rapamycin, but is highly, and differentially, responsive to treatment with this drug: increasing in healthy tendon, but decreasing in injured tissue. Rapamycin targets mTOR, a master metabolic regulator with complex involvement in cellular homeostasis; it is perhaps not unexpected that inhibition under different cellular states might result in opposing downstream alterations in expression. We also used our three high ranking references to normalize expression of *SDHA*, a gene which performed poorly in our overall dataset but markedly better in some subsets; as we show (**Fig 7**), expression of this component of the mitochondrial respiratory chain is indeed comparatively stable within groups, and indeed exhibits no clear rapamycin-associated changes, but is modestly but consistently reduced following injury. We also note that previous investigations into tendon aging have demonstrated that Advanced Glycation End-Products (AGEs, by-products which accumulate with ageing and injured tendons), suppress *SDHA* mRNA levels [40].

Our study is comprehensive, using suitable sample cohorts (N = 3–5) and an extensive panel of candidate references (11, demonstrably capable of both determining strong candidates and identifying clearly inappropriate genes), however we must acknowledge some limitations. All samples here are collected from female rats (males were omitted to reduce likelihood of post-surgical fighting within cages), rendering our data statistically well-powered for one sex only; we do not expect strong sexually dimorphic responses to either rapamycin or tendon injury, but we cannot discount this as a possibility [41]. Our reference panel is also not exhaustive: other candidates used in previous studies such as *RPLP0*, *PPIB*, *POLR2A* [39] might serve as even better references (though given the empirically strong scoring of *CSNK2A2*, *HPRT1* and *ACTB*, any further gains in stability would be minor at best). We also note that total RNA yields from uninjured (control) tendons were typically lower than from injured: as discussed above, healthy tendons are largely transcriptionally inert, while injured tendons are host to a diverse cellular milieu of active transcriptional turnover (as reflected in *18S* levels). Such differences in cell content and activity potentially render assessment of gene expression challenging, but our data nevertheless remain robust and well supported (with multiple different assessments identifying the same candidates), offering a viable solution to such challenges.

## Conclusions

We have investigated 11 candidate reference genes for normalizing qPCR data in an adult Wistar rat model of tendon injury, with or without rapamycin administration. Our data suggests that *ACTB*, *CSNK2A2*, *HPRT1*, and *PAK1IP1* are all strong references: any combination of these genes would be suitable for normalizing gene expression data in this complex comparative scenario. We show that two reference genes are sufficient, but use of three provides more powerful normalisation. We further demonstrate that other popular RGs are actively poor choices, with normalisation to three of our strong candidates (*ACTB*, *CSNK2A2*, and *HPRT1*) indicating that *GAPDH* and *SDHA* are injury and/or rapamycin sensitive and therefore should be avoided as RGs in future studies.

## Supporting information

**S1 Fig. Ungrouped analyses using NormFinder.** Rankings for 11 candidate RGs computed by Normfinder using ungrouped analyses of uninjured (**A**), injured (**B**), rapamycin treated (**C**), and vehicle treated tendons (**D**). Lower scores correspond to higher stability. Best pair combinations are indicated in boxes respectively.
(TIF)

**S1 Table. Pairwise variation output from GeNorm analyses.** Variability values of pairwise variation with increasing number of RGs for entire dataset (or subdata as indicated. Values $\leq 0.2$ are considered acceptable, therefore three RGs reduce variability but two (best-pair) suffice in all instances.
(CSV)

**S1 Raw data.**
(CSV)

## Author Contributions

**Conceptualization:** Neil Marr, Richard Meeson, Richard J. Piercy, John C. W. Hildyard, Chavaunne T. Thorpe.

**Data curation:** Neil Marr, John C. W. Hildyard.

**Formal analysis:** Neil Marr, John C. W. Hildyard.

**Funding acquisition:** Richard Meeson, Chavaunne T. Thorpe.

**Investigation:** Neil Marr, John C. W. Hildyard.

**Methodology:** Neil Marr, John C. W. Hildyard.

**Project administration:** Richard Meeson, Chavaunne T. Thorpe.

**Resources:** Neil Marr, Richard Meeson, John C. W. Hildyard, Chavaunne T. Thorpe.

**Software:** Neil Marr, John C. W. Hildyard.

**Supervision:** Richard Meeson, John C. W. Hildyard, Chavaunne T. Thorpe.

**Validation:** Neil Marr, John C. W. Hildyard.

**Visualization:** Neil Marr, John C. W. Hildyard.

**Writing – original draft:** Neil Marr, Richard Meeson, Richard J. Piercy, John C. W. Hildyard, Chavaunne T. Thorpe.

**Writing – review & editing:** Neil Marr, Richard Meeson, Richard J. Piercy, John C. W. Hildyard, Chavaunne T. Thorpe.

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
