## [Decision Letter · Decision Letter 0]

22 Jul 2024

PONE-D-24-25318

Evaluation of suitable reference genes for qPCR normalisation of gene expression in a Achilles tendon injury model.

PLOS ONE

Dear Dr. Marr,

Thank you for submitting your manuscript to PLOS ONE. After careful consideration, we feel that it has merit but does not fully meet PLOS ONE’s publication criteria as it currently stands. Therefore, we invite you to submit a revised version of the manuscript that addresses the points raised during the review process.

We look forward to receiving your revised manuscript.

Kind regards,

Zulkarnain Jaafar

Academic Editor

PLOS ONE

Journal Requirements:

 "Funded by Versus Arthritis (22607)."

Additional Editor Comments:

** Dear author, please make necessary changes based on the comments provided by the reviewers.**

Reviewers' comments:

Reviewer's Responses to Questions

**Comments to the Author**

1. Is the manuscript technically sound, and do the data support the conclusions?

Reviewer #1: Yes

Reviewer #2: Yes

2. Has the statistical analysis been performed appropriately and rigorously? 

Reviewer #1: Yes

Reviewer #2: Yes

3. Have the authors made all data underlying the findings in their manuscript fully available?

Reviewer #1: Yes

Reviewer #2: Yes

4. Is the manuscript presented in an intelligible fashion and written in standard English?

Reviewer #1: Yes

Reviewer #2: Yes

5. Review Comments to the Author

Reviewer #1: The text is well written and the authors have good arguments with the results obtained to defend the conclusion. I send some suggestions and corrections:

Table 2: remove brackets in sample number inside the table

Table 2 is cited before Table 1, the author should reverse the order of the tables and check the entire order of the tables and figures.

I suggest adding in the legend of Fig 1 that lower Cq values indicate greater gene expression

Some references are cited in the wrong order, line 293 refs 36-38, then on line 308 appears ref 29 followed by 28, line 311 refs 26 and 27. Check the rest of the text.

Reviewer #2: The manuscript by Neil Marr etc. proved that ACTB, CSNK2A2, HPRT1 and PAK1IP1 are all stably expressed in tendon, regardless of injury or drug treatment: any three of these would serve as universally suitable reference gene panel for normalizing qPCR expression data in the rat tendon injury model. And 18S, UBC, GAPDH, and SDHA as consistently poor scoring candidates. They should be avoided. This article is well written and designed. I have only a few minor concerns：

1.How did the author determine the reference candidates?

2.Lines 170-172, 181-186：please indicate the results

3.Please introduce the results separately for A, B, and C in each figure

4.How did calculate the expression of target genes using ACTB, HPRT1, and CSNK2A2 as a reference gene panel? Calculate the target gene expression using the geometric mean of Cq values for ACTB, HPRT1, and CSNK2A2?

6. PLOS authors have the option to publish the peer review history of their article (what does this mean?). If published, this will include your full peer review and any attached files.

Reviewer #1: No

Reviewer #2: No

---

## [Author Response · Author response to Decision Letter 0]

5 Aug 2024

Dear Editor,

We would like to thank the reviewers for their constructive comments and feedback: we have revised our manuscript accordingly and agree that the suggested changes are both sensible and well-judged. As requested, please find our responses to each point raised by both editor and reviewers below.

Editor comment 1:

“When submitting your revision, we need you to address these additional requirements.

https://journals.plos.org/plosone/s/file?id=ba62/PLOSOne_formatting_sample_title_authors_affiliations.pdf”

Response to Comment:

We have reviewed and changed all non-compliant styles within our manuscript with specific attention to formatting styles of file naming, main body text and title/author affiliations. 

Editor comment 2:

“To comply with PLOS ONE submissions requirements, in your Methods section, please provide additional information regarding the experiments involving animals and ensure you have included details on (1) methods of sacrifice, (2) methods of anesthesia and/or analgesia, and (3) efforts to alleviate suffering.” 

Response to Comment:

We have now provided additional information regarding the experimentation, pre- and post-operative analgesia, euthanasia, and observations performed to alleviate suffering / humane endpoints (lines 111-117, line 125). 

Editor comment 3:

“Please note that funding information should not appear in any section or other areas of your manuscript. We will only publish funding information present in the Funding Statement section of the online submission form. Please remove any funding-related text from the manuscript.”

 Response to Comment:

We have removed all funding related test from the manuscript.

Editor comment 4:

“Thank you for stating the following financial disclosure: 

 "Funded by Versus Arthritis (22607)."

Please include this amended Role of Funder statement in your cover letter; we will change the online submission form on your behalf.”

Response to Comment:

We thank the Editor for providing a Role of Funders statement; the recommended statement accurately reflects the role of our funders, and therefore we have included this statement in our revised cover letter.

Editor comment 5:

“Please review your reference list to ensure that it is complete and correct. If you have cited papers that have been retracted, please include the rationale for doing so in the manuscript text, or remove these references and replace them with relevant current references. Any changes to the reference list should be mentioned in the rebuttal letter that accompanies your revised manuscript. If you need to cite a retracted article, indicate the article’s retracted status in the References list and also include a citation and full reference for the retraction notice.”

Response to Comment:

We have reviewed the reference list and have not identified any retracted papers. We can confirm that the reference list is complete and correct.

Additional Editor Comments:

“Dear author, please make necessary changes based on the comments provided by the reviewers.”

Response to Comment:

We have implemented all changes requested by the reviewers: please see point by point responses below.

Reviewer 1 comment 1:

“The text is well written and the authors have good arguments with the results obtained to defend the conclusion. I send some suggestions and corrections:

Table 2: remove brackets in sample number inside the table

Table 2 is cited before Table 1, the author should reverse the order of the tables and check the entire order of the tables and figures.”

Response to Comment:

We thank the reviewer for their positive comments on our manuscript, and for spotting the incorrect numbering. We have (1) corrected the order of tables referenced in the manuscript, and (2) removed all brackets as suggested within Table 1.

Reviewer 1 comment 2:

“I suggest adding in the legend of Fig 1 that lower Cq values indicate greater gene expression”

Response to Comment:

A sensible suggestion: thank you. We recognise the relationship between Cq and expression level is counter-intuitive, and have accordingly now included a statement (lines 186-187) to better inform readers.

Reviewer 1 comment 3:

“Some references are cited in the wrong order, line 293 refs 36-38, then on line 308 appears ref 29 followed by 28, line 311 refs 26 and 27. Check the rest of the text.”

Response to Comment:

We appreciate the reviewer’s attention to detail. We can confirm the numbering is correct as currently presented: references 26-29 are also cited earlier (within our introduction on line 95).

 

Reviewer 2 comment 1:

“The manuscript by Neil Marr etc. proved that ACTB, CSNK2A2, HPRT1 and PAK1IP1 are all stably expressed in tendon, regardless of injury or drug treatment: any three of these would serve as universally suitable reference gene panel for normalizing qPCR expression data in the rat tendon injury model. And 18S, UBC, GAPDH, and SDHA as consistently poor scoring candidates. They should be avoided. This article is well written and designed. I have only a few minor concerns：

1.How did the author determine the reference candidates?”

Response to Comment:

We thank the reviewer for their positive review. As our study clearly shows, traditional reference gene candidates (GAPDH, 18S, ACTB) are not appropriate under all scenarios, and indeed can be actively detrimental to good normalisation. Our candidate references are adapted primarily from the GeNorm and GeNormPLUS gene lists devised by Primerdesign: a series of ostensibly stable genes identified in unbiased fashion via datamining of public repository gene expression data (chiefly microarray), intended for use in exactly the type of study presented here. Within these lists, the three candidates listed above are typically included for comparative purposes, allowing investigators to empirically assess the performance of strong candidates against these traditional three (which often perform poorly). In some instances (as here) one or more of these candidates performs well, a finding of considerable value.

Some of these genes (i.e. RPL13a, HPRT1) have been shown by other groups to be strong references in many model systems, and we have found this panel to be capable of both determining strong references and also identifying actively poor references, or references that are highly context-specific. We have used these previously in cell culture, tissue and animal models, using both mouse and canine samples: citations of these earlier works (which more comprehensively address the rationale outlined above) are included within the text (lines 92-95; refs 23-29). 

We note that this work, in a different rodent model (the rat) required rat-specific primers: these were either taken from the literature (where appropriate), or designed de novo to rat sequence (as described in the text).

Reviewer 2 comment 2:

“2.Lines 170-172, 181-186：please indicate the results”

Response to Comment:

The authors have now indicated results in the suggested lines as well as throughout the manuscript (lines 192, 195, 198, 213-215, 217-219, 239-240, 245-246, 260-261, 273-274, 290-291).

Reviewer 2 comment 3:

“3.Please introduce the results separately for A, B, and C in each figure”

Response to Comment:

We now introduce each individual subfigure as requested, throughout the manuscript: in some places we have necessarily made minor edits to the text to facilitate this (lines 192, 195, 198, 213-215, 217-219, 239-240, 245-246, 260-261, 273-274, 290-291).

Reviewer 2 comment 4:

“How did calculate the expression of target genes using ACTB, HPRT1, and CSNK2A2 as a reference gene panel? Calculate the target gene expression using the geometric mean of Cq values for ACTB, HPRT1, and CSNK2A2?”

Response to Comment:

We thank the Reviewer for raising this point; we in fact used the geometric mean of relative quantity (RQ) values for ACTB1, HPRT1 and CSNK2A2. In detail: all Cq values for all genes were first converted to RQ -as noted in the text, this is necessary for geNorm and Normfinder analysis. Our three gene normalisation factor (NF) was the geometric mean of per-sample RQ values for ACTB1, HPRT1 and CSNK2A2 (i.e. reference genes universally suitable for uninjured or injured tendon treated with or without rapamycin), and normalisation was accordingly conducted conventionally (GOI RQ divided by NF). Resultant normalised values (or equivalent non-normalised RQ data) were then log transformed for statistical analysis. Mathematically, with reaction efficiencies of 100% this is essentially equivalent to -dCt, and indeed we accept that many investigators choose to remain within log-space (Cq values) from data collection to analysis. Our approach however allows per-reaction efficiency to be incorporated where necessary, and minimises the risk of uncoupling the derived values from the underlying biological data: “-3.94 vs -1.17” is less immediately intuitive than ‘6.8 vs 1”. We find that converting data to linear-space values facilitates assessment of overall data distribution and affords a greater appreciation for the magnitude of fold-changes involved (and readily permits outlier identification, where necessary): we have used this approach in prior work (cited within this manuscript), for these reasons, and we use it here for the same reasons, and for consistency with published literature. We have added a short explanation to the manuscript text to better communicate our methodology to the reader (lines 167-170). 

Lastly, we once again thank the editor and reviewers for their valuable feedback and constructive criticism of our manuscript.

Yours Sincerely,

Dr Neil Marr

Comparative Biomedical Science

The Royal Veterinary College

Royal College Street

London

NW1 0TU

United Kingdom

---

## [Decision Letter · Decision Letter 1]

13 Aug 2024

Evaluation of suitable reference genes for qPCR normalisation of gene expression in a Achilles tendon injury model.

PONE-D-24-25318R1

Dear Dr. Marr,

We’re pleased to inform you that your manuscript has been judged scientifically suitable for publication and will be formally accepted for publication once it meets all outstanding technical requirements.

Kind regards,

Zulkarnain Jaafar

Academic Editor

PLOS ONE

Additional Editor Comments (optional):

Reviewers' comments:

Reviewer's Responses to Questions

**Comments to the Author**

1. If the authors have adequately addressed your comments raised in a previous round of review and you feel that this manuscript is now acceptable for publication, you may indicate that here to bypass the “Comments to the Author” section, enter your conflict of interest statement in the “Confidential to Editor” section, and submit your "Accept" recommendation.

Reviewer #1: All comments have been addressed

Reviewer #2: (No Response)

2. Is the manuscript technically sound, and do the data support the conclusions?

Reviewer #1: Yes

Reviewer #2: (No Response)

3. Has the statistical analysis been performed appropriately and rigorously? 

Reviewer #1: Yes

Reviewer #2: (No Response)

4. Have the authors made all data underlying the findings in their manuscript fully available?

Reviewer #1: Yes

Reviewer #2: (No Response)

5. Is the manuscript presented in an intelligible fashion and written in standard English?

Reviewer #1: Yes

Reviewer #2: (No Response)

6. Review Comments to the Author

Reviewer #1: The authors addressed the recommended changes and I believe that the manuscript is ready for publication

Reviewer #2: (No Response)

7. PLOS authors have the option to publish the peer review history of their article (what does this mean?). If published, this will include your full peer review and any attached files.

Reviewer #1: No

Reviewer #2: No

---

## [Editor Report · Acceptance letter]

16 Aug 2024

PONE-D-24-25318R1 

PLOS ONE

Dear Dr. Marr, 

I'm pleased to inform you that your manuscript has been deemed suitable for publication in PLOS ONE. Congratulations! Your manuscript is now being handed over to our production team.

Kind regards, 

on behalf of

Dr. Zulkarnain Jaafar 

Academic Editor

PLOS ONE